# Thermal Behavior of Passive Intelligent Radiant Cooling Systems

Seung-Ho Yoo

Solar Architecture Laboratory, Sehan University, Youngam 58447, Republic of Korea; mryooo@naver.com

**Abstract:** Efficient cooling and heating solutions for nearly zero-energy solar dwellings are required to mitigate climate change and to make dwellings sustainable. The installed pipeline for a radiant heating system, which is only used for space heating when heating is necessary, can also be used to cool the room with only the enthalpic use of natural city water by releasing the natural city water through the embedded pipeline already installed for radiant heating. Natural city water used for radiant cooling can be used in necessary locations such as for toilets, washing cars, laundry facilities, and garden water, which corresponds to approximately 56% of the water we use at home. As a result, the embedded pipes that make up a radiant heating system can be converted to a passive intelligent radiant cooling system with minimal added installation and control systems. Thermal comfort and behavior analyses in an enclosure with a radiant cooling system are fulfilled through experimentation, mean radiant temperature simulation, and asymmetric radiation calculation. No uncomfortable asymmetric radiation is encountered during the cooling period, so the cooling spaces are well controlled within the comfortable cooling range. A passive intelligent radiant cooling system that uses just the enthalpy of natural city water can be an appropriate ecological solution to better develop zero-energy dwellings. No extra cooling energy and power are required to cool a space that uses just enthalpy and pressure from natural city water.

**Keywords:** radiant cooling; no energy input; On-dol heating; mean radiant temperature; natural water; thermal comfort; asymmetric radiation; enthalpy

## 1. Introduction

Buildings are critical to the transition to a net-zero future, as they are responsible for about 40% of global energy consumption and about one-third of global GHG emissions. Energy consumption for space cooling has more than tripled since 1990, especially during peak demand periods and extreme heat events. Over 10% of the building energy used is applied to air conditioning and indoor thermal comfort in hot seasons. Changing the air conditioning mode is one solution to meet the cooling demand without increasing power consumption and $CO_2$ emission. Global space cooling demand continued to grow in 2020, driven in part by greater home cooling as more people spent time at home. Space cooling accounted for nearly 16% of the building sector's final electricity consumption in 2020. Residential AC units in operation for space cooling account for nearly 70% of the total [1–3].

An environmentally friendly or energy-efficient heating and cooling system attracts great attention due to energy, environmental problems, and climate change, etc. Radiant heating and cooling systems are ideal examples of these cases.

The balance of heat within the human body is about 46–50% influenced by radiation exchange in the built environment [4,5]. Therefore, thermal characteristics in a radiant built environment need to be precisely accessed through an efficient evaluation method [6]. The heat flow density on a floor surface was determined by pipe spacing, thickness, and heat conductivity of the layer above the pipe, etc. [7]. Radiant cooling can be an energy-efficient strategy, thereby reducing both sensible and latent loads in spaces [8].

In Germany, the first reports have appeared on the application of and experimentation with ceiling systems that are heated in the winter and used for cooling purposes at the

same time in the summer [9]. In Switzerland, a thermal active building structure system has been utilized for radiant heating and radiant cooling purposes [10].

Most Korean dwellings have conventionally used an On-dol (warm stone plate) heating system, a traditional radiant floor heating system. A radiant cooling system attracts a lot of attention nowadays from the viewpoint of energy conservation and thermal comfort.

The number of articles investigating radiative cooling structures for building applications has increased rapidly in recent years [11]. In air-based cooling systems that consist of a thermal radiator, the cooled air is either circulated artificially by a fan or by a natural process due to the buoyancy effect. That is, in these systems, the air is used as the heat exchange medium that is directly heated by the interior environment and cooled by the thermal radiator. Similarly, in water-based cooling systems, water is utilized as a medium to transfer heat. These systems normally consist of a thermal radiator, insulated water tank, heat exchanger, and water pump. Water is cooled by the thermal radiator at night and then stored in an insulated water tank. During the day, the cooled water is circulated with the help of a water pump through the heat exchanger to provide space cooling. In hybrid systems, chilled water provided by a radiative cooler is utilized for the cooling coils in an air-conditioning system to enhance the system's efficiency [11].

This study uses a water-based system in which the low temperature of city water is directly utilized as a medium to provide space radiant cooling. This system does not need a radiator, fan, water tank, heat exchanger, water pump, cooling coil, or chiller.

Thermal behaviors, such as comfort characteristics and asymmetric radiation in an enclosure with passive intelligent radiant cooling and the thermal capacity of the system in a dwelling, are analyzed by experiments and simulations through their application in a private house.

## 2. Materials and Methods

### 2.1. Principle of Passive Intelligent Radiant Cooling

Conventional radiant heating systems, which are used for space heating only in cooler temperatures, can also be used to cool a room without causing any loss or waste of city water or power input by releasing city water again through an embedded pipeline already installed in conventional radiant heating systems while the occupants are using the water at home. This is a form of the passive intelligent radiant cooling concept that is controlled and cooled by just the water pressure and enthalpy of the city water, like tap water. If we open the water tap, water is automatically released by the pressure of the city water. In Korea, even when the outside air temperature is over 36 °C during the cooling period, natural city water has a temperature of approximately 19–20 °C in principle because the city water is supplied at least 90 cm under the earth, according to Korean law. The corresponding city water temperatures are, respectively, 19–20 °C in Turkey and 10–14 °C in Germany and Denmark [12]. Natural city water flows through a pipeline that is embedded in the cooling surface during the occupants' water use for toilets, washing machines, cleaning, car washing, garden watering, and showering, which corresponds to approximately 56% of the water we use at home [13]. It is similar to one of the multi-purpose principles in which the room is automatically heated by the hot exhaust that flows into the heating flue after burning timbers to cook something in a kitchen in a traditional On-dol (warm stone plate) radiant heating system in Korea [14,15]. No power or water is required to cool the room. The enthalpy of natural city water for radiant cooling is used to cool the room space, and the used water continues to flow through the system for necessary tasks such as using toilets, running washing machines, washing cars, cleaning, and watering gardens.

Figure 1 shows the principles of the radiant cooling system without any energy input [16].

### 2.2. Description of the Passive Intelligent Radiant Cooling System

The thermal behavior and capacity of the passive intelligent radiant cooling system are evaluated in a house located in Chung-Nam Province, Republic of Korea. Figure 2

shows the section of the attached glass house (Figure 2a,b) and the floor plan of a radiant cooling and heating system (Figure 2c).

The radiant floor heating system was originally installed under the floor of a private house. The blue colored arrows show the final flow vector of the water heated through thermal heat exchange with the concrete floor while passing pipelines installed in rooms a, b, and the glass house.

The total dimensions of the house are approximately 11.7 m × 8.7 m × 2.5 m (Horizontal axis × Vertical axis × Height), including the glass house, which is attached. The glass is made up of vacuum glazing, and the outer wall has been remodeled with 5 cm thick sandwich paneling, but the interior wall remains the original 10 cm thick traditional earthen wall. The glass window between the glass house and room a is 3 mm thick.

Figure 3 shows a section of the radiant cooling and heating pipe installation plan.

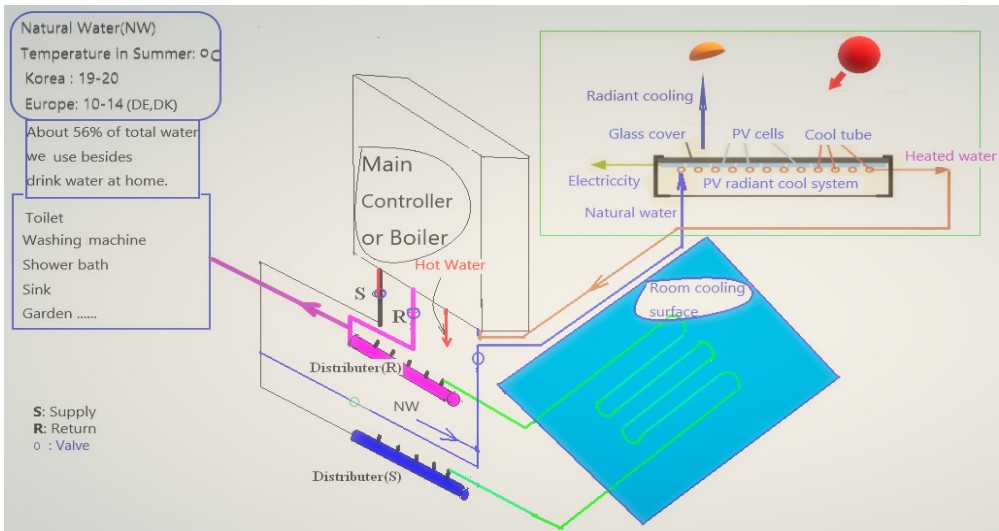

**Figure 1.** Principles of radiant cooling without any energy input.

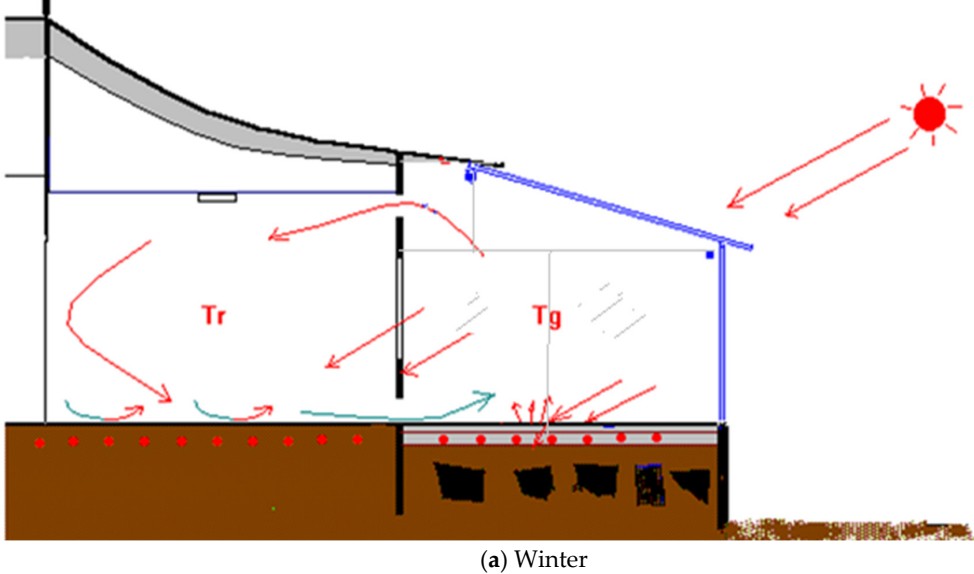

(**a**) Winter

**Figure 2.** *Cont*.

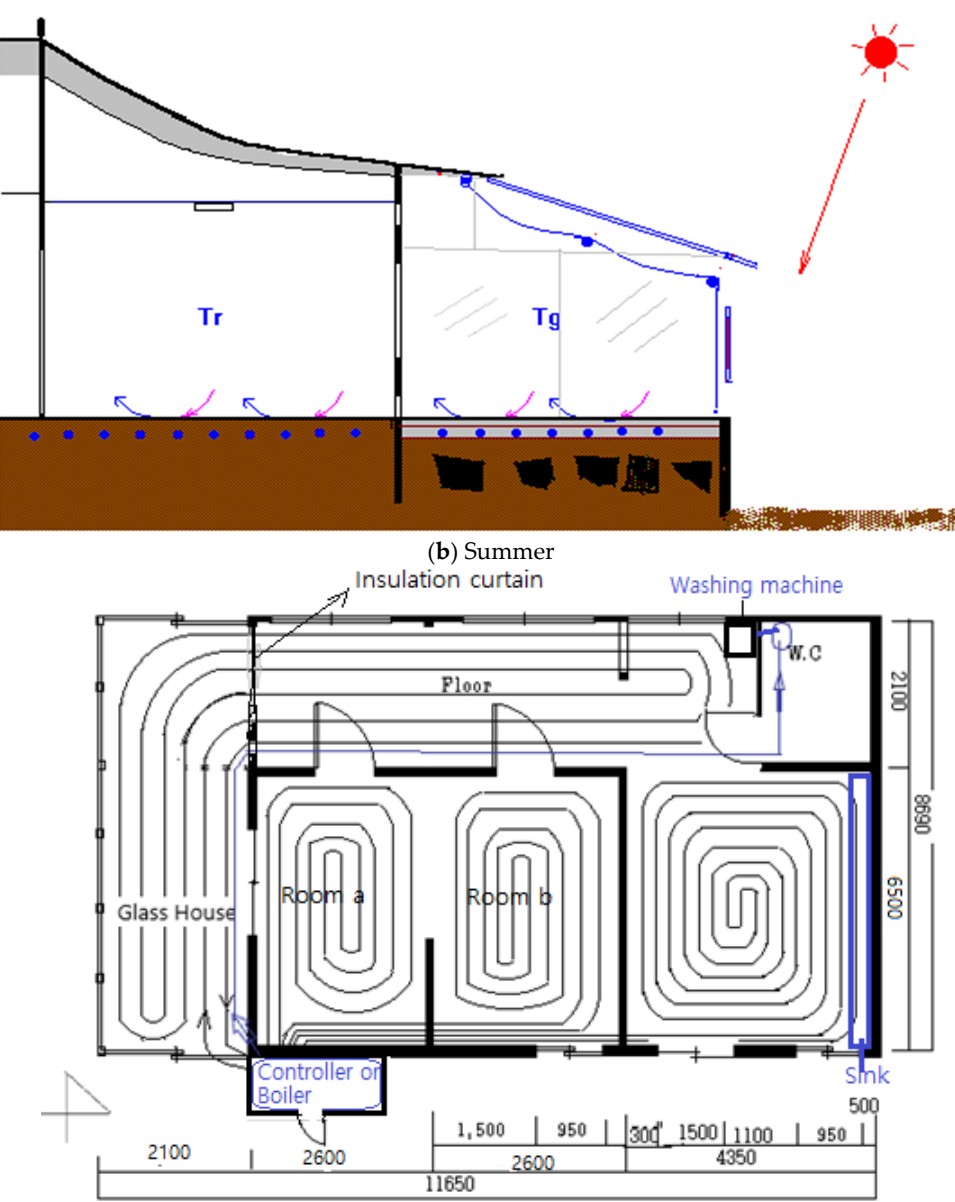

(**b**) Summer

(**c**) Floor plan of a radiant cooling and heating system

**Figure 2.** Section of attached glass house (**a**,**b**) and floor plan of a radiant cooling and heating system (**c**).

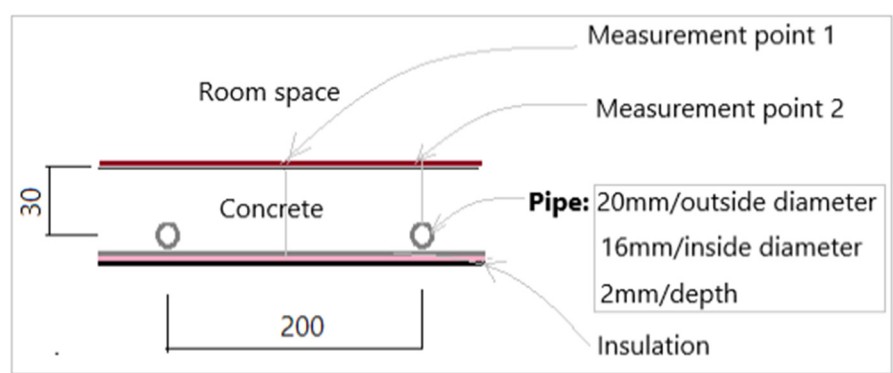

**Figure 3.** The section of the radiant cooling and heating pipe installation plan.

The material for the radiant cooling pipe is polyethylene plastic. The distance between pipes is 200 mm, and the depth of the concrete from the pipe to the floor surface is 30 mm.

The house was originally constructed with an On-dol (warm stone plate) radiant heating system, which is heated by hot gas through a timber combustion system when cooking in a kitchen and has been in use since approximately the end of the 1800s. The On-dol radiant heating system was converted to the present hot water On-dol radiant heating system in the early 1980s. Similarly, almost all such traditional On-dol radiant heating systems have been improved to a radiant heating system heated by the hot water that flows through the pipeline embedded in the concrete flooring. The hot water is circulated through the embedded pipeline to heat the room by the boiler in winter.

The glass house has been attached since the late 1990s. The target passive intelligent radiant cooling system was converted in 2018 to cool the house without any energy and power input.

Even when the outside air temperature is over 36 °C during the cooling period, the natural city water is approximately 19–20 °C in the Republic of Korea. In summer, the natural city water flows to cool the room through a pipeline embedded in the floor surface. No power and no extra water are requested to cool the room space. The more people who live in the radiant cooling space, the more city water automatically flows to cool the room space by the water pressure of city water because more water is used as the number of occupants increases.

### 2.3. Research Methodologies

Thirty-two thermal points have been measured by the data logger every thirty minutes, including the vertical room air, the surface temperature of the room, and the other boundary conditions, including the inlet and outlet water temperature for the room and the outdoor air temperature. Tg is the glass house temperature. Tr is the room temperature. Table 1 shows the measurement details.

**Table 1.** Measurement details.

| Measurement Factor (Position) | Method | Accuracy |
|---|---|---|
| Outdoor air temperature (1.5 m) | | |
| Vertical room temperature (Floor surface, 0.1 m, 0.8 m, 2.3 m, and ceiling surface) | | |
| Vertical floor temperature (Floor surface, 0.15 m, 0.8 m) | Data logger (Datascan 7020) + T-type thermocouple | |
| Inside surface temperature (South wall, south window, west wall, north wall, floor surface 1, 2, 3, 4) | | ±0.75% |
| Inside temperature of the glass house -Surface (Floor, south, east, and inclined glass ceiling) -Vertical air (0.1 m, 0.8 m, 2.3 m) - Inlet and outlet water temperature (Pipe surface temp.) | | |
| Glove temperature (Room and glasshouse) (0.8 m) | Glove thermometer + T-type thermocouple | |
| Room humidity (0.15 m) | Asman humidity meter + T-type thermocouple | |

Some simulations to figure out the thermal behavior of the system are fulfilled by the use of measurement data such as air temperature, surface temperature, and inlet and outlet water temperature to calculate the mean radiant temperature, operative temperature, and thermal capacity of the radiant cooling system.

### 3. Thermal Behavior of the Radiant Cooling System

### 3.1. Evaluation of Mean Radiant Temperature (MRT) and Operative Temperature (OT)

A cubic box model, which has the same surface area as the human body, is used to evaluate thermal comfort characteristics such as MRT and OT.

The human body surface area $A_{du}$ is calculated according to the following Equation (1).

$$A_{du} = 0.203 \times W^{0.425} \times H^{coef} \tag{1}$$

Herein, the coefficient coef is 0.696 for a sitting person. The weight W of a person is 70 kg, and height H is 110 cm for a sitting person. The MRT is calculated by the following Equation (2).

$$\text{MRT} = (\sum \emptyset_{b,si} \times T_{si}{}^4)^{1/4} \tag{2}$$

Herein, $\emptyset_{b,si}$ is the view factor between the human body model and room surface, and $T_{si}{}^4$ is the absolute temperature of the surrounding surface, s.

The operative temperature $t_{ot}$ is calculated by the following Equation (3) [17].

$$t_{ot} = a \times t_{air} + b \times \text{MRT} \tag{3}$$

Herein a + b = 1, a: 0.5, and b: 0.5.

### 3.2. Asymmetric Radiation

A small cubic box of 1 cm$^2$ on a height of 0.6m is considered to calculate the asymmetric radiation among the room surfaces, including the floor cooling surface.

The asymmetric radiation temperature (Asym) is calculated by the following Equation (4) [17].

$$\text{Asym} = |\text{MRT}_u - \text{MRT}_o| \tag{4}$$

Herein, $\text{MRT}_u$ is the MRT for the underside, and $\text{MRT}_o$ is the MRT for the opposite side.

The simulation program "COMFORT" is used to calculate the MRT, operative temperature (OT), and asymmetric radiation temperature, which can evaluate thermal behavior, including thermal comfort characteristics for the target house [18].

### 3.3. Cooling Capacity

The cooling capacity of the passive radiant cooling system is calculated according to Equation (5)

$$Q = m \times (h_2 - h_1) \tag{5}$$

Herein, $h_2 - h_1 = 4.186 \times (t_2 - t_1)$ in kJ/kg, m is the water flow rate (kg/s), $t_1$ is the inlet temperature, and $t_2$ is the outlet temperature.

Equation (5) is proportional to the water flow rate and inlet and outlet temperature difference [19].

## 4. Experiment and Simulation Results

### 4.1. Radiant Cooling Concept and Experiment Results for the Target House

A radiant heating system, which is only used for space heating during times when heating is necessary, can also be used to cool a room by releasing the natural city water through the embedded pipeline already installed for the radiant heating system. The city water used for radiant cooling can also be used for necessary household tasks such as flushing toilets, washing cars, using washing machines, and watering gardens, etc. For this purpose, even when the outside air temperature is maintained over 35 °C during summer, the city water is approximately 19–20 °C in the Republic of Korea, 10 °C (Germany) for tap water, so that the cooling surface flows through a pipeline embedded in the floor, wall, or ceiling where it is not needed for drinking water, whenever the occupants use the water system. No power is needed for the water to flow, and no wastewater is produced, either. The more people who live in a radiant cooling room, the more city water automatically flows to cool the room space by the water pressure of city water. The amount of city water flow was approximately 0.112–0.115 kg/s for 4 spaces (the glasshouse, rooms a and b, and the living room) during periods when regular functions like using washing machines and toilets and cleaning and showering were done while the occupants were at home.

Figure 4 shows the temperature variations during the radiant cooling operation depending on the time-sequential process.

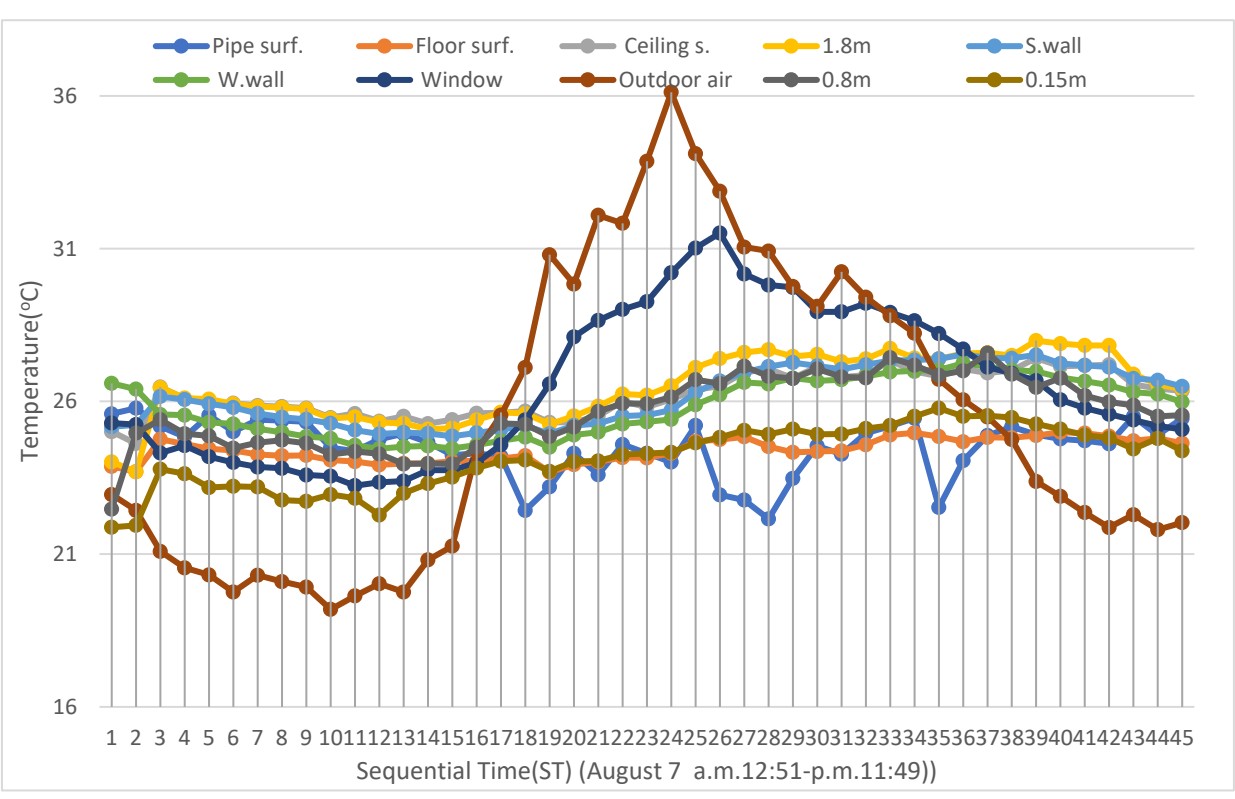

**Figure 4.** Temperature variations of the radiant cooling space depend on the time-sequential process.

Measurements were made every 30 min starting from 12:45 am (a.m. 12:51–p.m. 11:49) for 24 h. The highest outdoor air temperature is 36.1 °C at 12:51 (ST (Sequential time): 24). The gardening water was supplied from a.m. 09:30–09:45. The sequential time corresponds to 17–18 (a.m. 09:12–09:44) in Figure 4. A person took a shower from a.m. 11:00–11:12 (21 (ST): 11:17:36). The washing machine was used from p.m. 13:35–15:25 (25 (ST): p.m. 1:22:51, about 75 L water consumption per one load of laundry). The gardening and car washing were done from 18:05–18:33 (34 (ST): p.m. 6:04:38). A person took a shower from 21:00–21:10 (40 (ST): p.m. 9:12:31). A person usually used the WC 8–9 times per day. If water is used during the shower, the laundry, and the WC usage, etc., the embedded pipe surface temperature automatically goes down so that the concrete structure that contacts the cooling pipe falls in temperature to cool the corresponding room. This causes a fall in the surrounding surface temperature, including the ceilings, walls, and window surfaces, increasing the thermal comfort for the occupant due to cold radiant effects. During laundry cycles, the floor surface temperature is 24.69 °C while the temperature at 0.15 m height is 24.64 °C. The ceiling surface temperature is 26.4 °C while the air temperature is 27.1 °C at the near position of the ceiling surface. The west wall surface, south wall surface, south window surface t, and air temperatures at a height of 0.8 m are, respectively, 26.22 °C, 26.67 °C, 31.51 °C, and 26.71 °C. The temperature difference profile between the surface temperature and air temperature at the same height shows a similar distribution to the temperature profile between the floor surface and 0.15 m height air temperature due to the cold radiation effect. However, the temperature difference profiles near the vertical surfaces and ceiling surface show, respectively, 0.7 °C (ceiling: 27.1–26.4), 0.5 °C (west wall: 26.71–26.22), 0.04 °C (south wall: 26.71–26.67), and −4.8 °C (window: 26.71–31.51). The similar temperatures at the south wall and window come from the poor insulation of the south wall and the window and the high air temperature of the glass house.

Figure 5 shows the temperature variations of the radiant cooling system depending on the time-sequential process

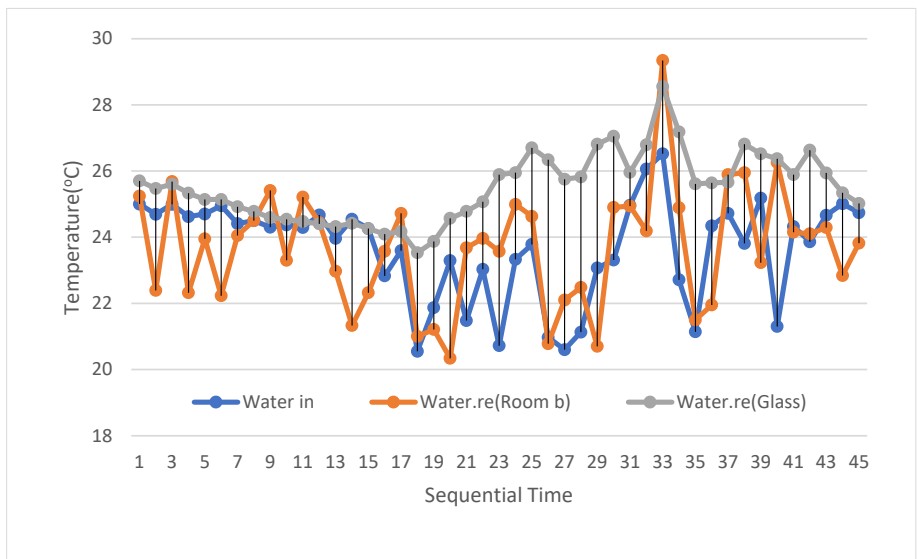

**Figure 5.** Temperature variations of the radiant cooling system depend on the time-sequential process.

The maximum outdoor air temperature was 36.12 °C at p.m. 12:51. Measurements of the glass house were collected for 1 h and 30 min from p.m. 12:51. The returned water was shown to be the highest temperature in the glasshouse. This demonstrates that the returned water (26–27 °C) from the glasshouse can be supplied to the boiler to make hot water in the summer to conserve energy. The inlet (city) water temperature was approximately 19–20 °C.

Figure 6 shows the average vertical temperature profile for the room for one day.

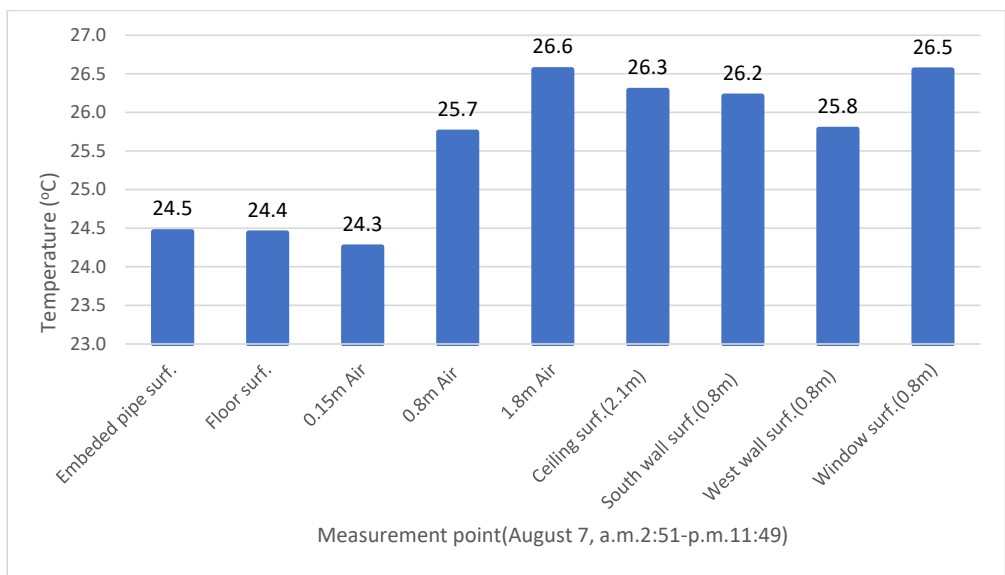

**Figure 6.** Vertical average temperature profile for the room for one day.

The average outdoor air and glass house temperatures for one day are, respectively, 25.4 °C and 27.5 °C.

The average room air temperature at a height of 0.15 m is the lowest at 24.3 °C, which could be comfortable for the occupants. An interesting fact is that the temperature at 1.8 m showed the ceiling surface temperature is 0.3 °C lower than the air temperature near the ceiling due to a radiant effect mainly from the floor's radiant cooling surface.

Figure 7 shows the vertical temperature profile for the room at p.m. 12:51.

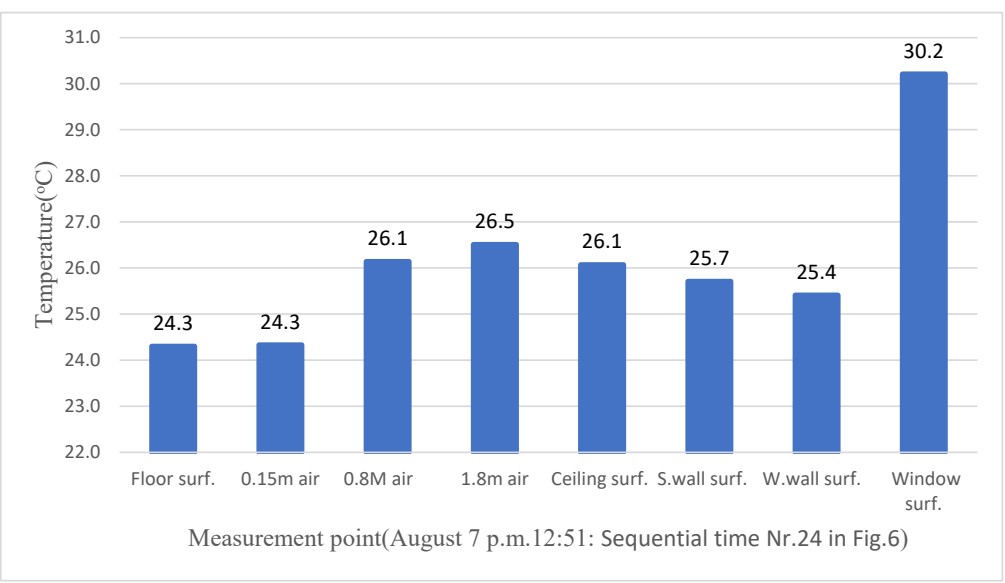

**Figure 7.** Vertical temperature profile for the room for one instant.

At this moment, the radiant cooling floor surface temperature is similar to the air temperature at 0.15 m height (near the floor cooling surface), so there is little convective heat transfer on the floor radiant cooling surface. Alternatively, the radiant cooling floor surface radiates a cold radiation flux to the ceiling surface, south wall surface, west wall surface, and window surface causing a fall in surface temperature. Therefore, the surface temperature of the ceiling and the west wall at 0.8 m height is lower or similar to the air temperature at 0.8 m height. However, the inside surface temperature of the window is 4.1 °C, which is higher than the air temperature at a height of 0.8 due to the high temperature (35.5 °C) of the adjacent glasshouse.

### 4.2. Mean Radiant Temperature (MRT) and Operative Temperature (OT) Calculation

MRT and OT are calculated by the simulation program COMFORT based on Equations (2) and (3).

Figure 8 shows the MRT distribution for the radiant floor cooling system for the house.

The total of 24 points, which keep a 1m distance from each of the 4 different room surface orientations, were chosen to simulate the MRT and operative temperature. The average temperature of the MRT was 25.27 °C, but the temperature at the window on one side was relatively higher than the inner side due to the high window surface temperature from the adjacent glass house. However, the effect of the floor surface temperature on the MRT is the most influential factor, and the average floor surface temperature of 24.4 °C is within the lowest percentage dissatisfied range as per the local thermal discomfort caused by warm or cold floors [17,19–21].

Figure 9 shows the operative temperature distribution for the radiant floor cooling space for the house.

The average operative temperature was 25.69 °C, which is a comfortable range for occupants. The room OT range is 25.5–25.8 °C even on the hottest day, but the temperature at the south window side was relatively higher than the inner side due to the higher south window surface temperature. The b of Equation (3) corresponding to the coefficient of the MRT should be more precisely applied for this kind of radiant cooling enclosure within a range that is greater than 0.5 and less than 1.0 because the influence of a radiant surface on an occupant's comfort is greater than with an air conditioning system.

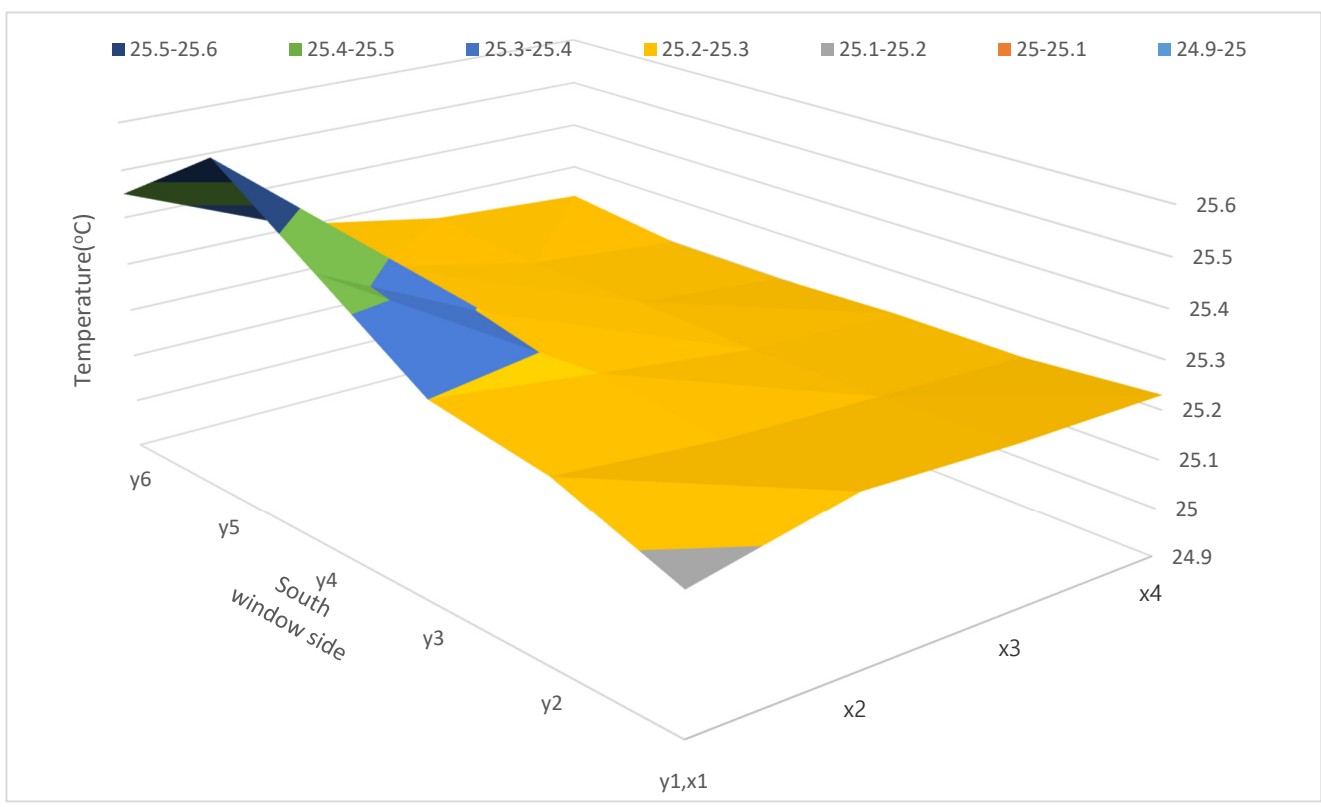

**Figure 8.** MRT distribution for the radiant floor cooling system for the house (7 August, p.m. 12:51; Outdoor air temperature: 36.1 °C).

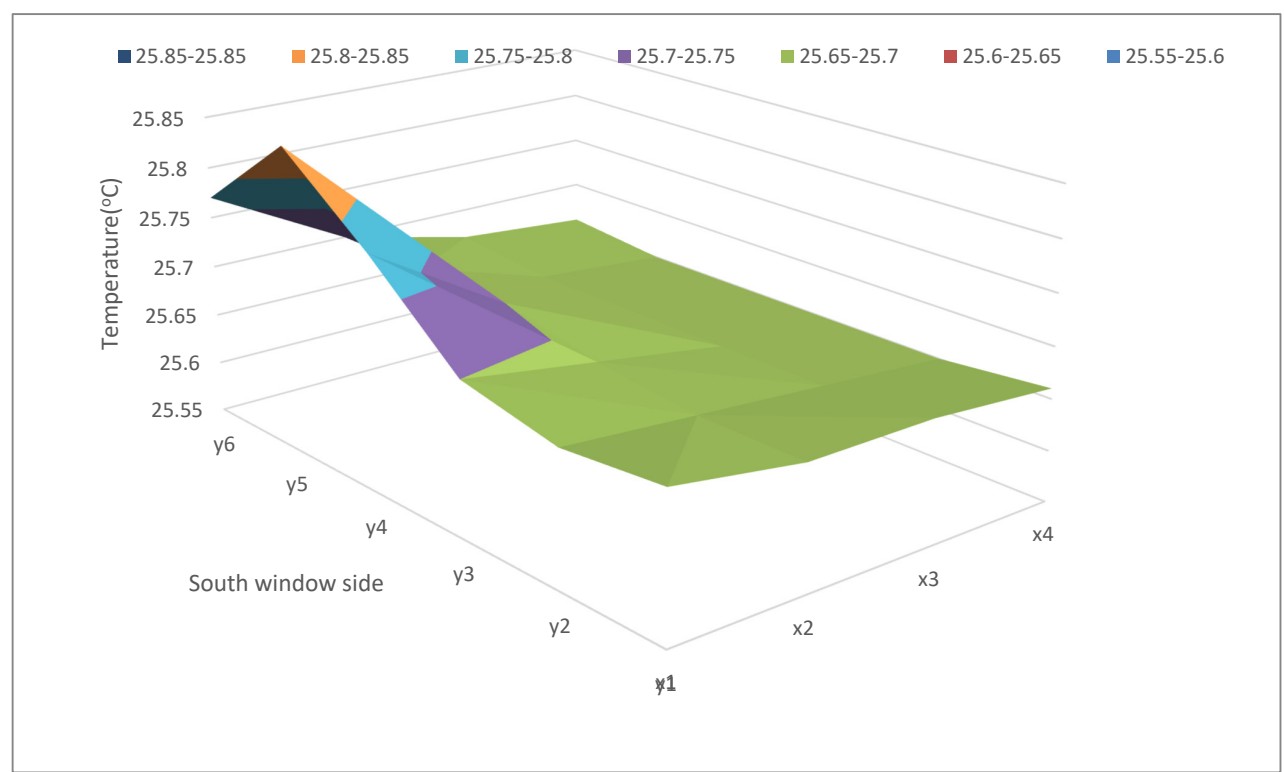

**Figure 9.** Operative temperature distribution for the radiant floor cooling space for the house (7 August, p.m. 12:51; Outdoor air temperature: 36.1 °C).

### 4.3. Asymmetric Radiation Simulation

The asymmetric radiation simulation for 6 different positions, which keep a 1 m distance from the south window and wall, was fulfilled by the use of the simulation program COMFORT based on Equation (4). The biggest possible temperature difference orientations between two different opposite surfaces were chosen to calculate the radiant asymmetry, namely the ceiling and floor surface, south window side, north sidewall surface, and east and west walls.

Figure 10 shows the calculation results for asymmetric radiation on 7 August at p.m. 12:51 (Sequential time Nr.24 in Figure 6)

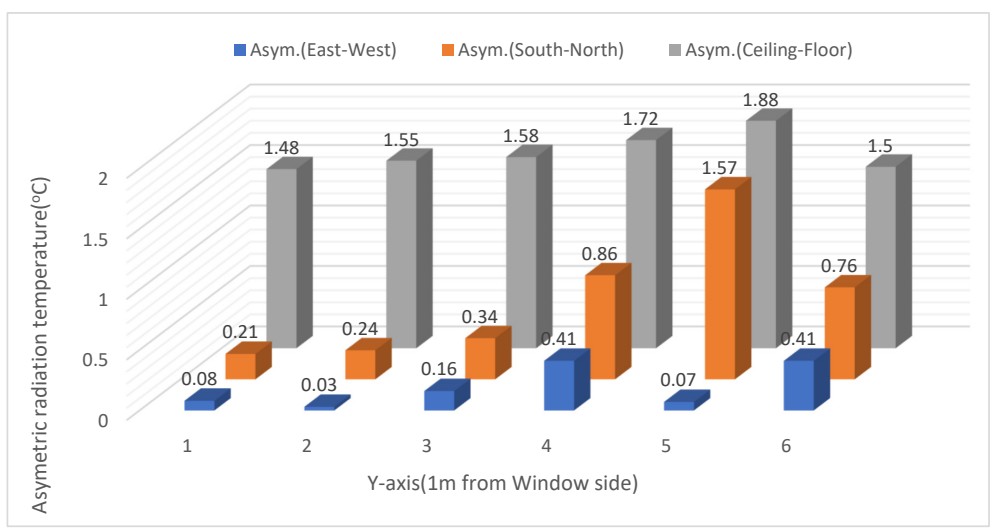

**Figure 10.** Asymmetric radiation (7 August, p.m. 12:51).

The maximum asymmetric radiation (1.88 °C) is raised at y5 (the ceiling and floor on the window side). Asymmetric radiation does not happen in this radiant cooling space because the floor's radiant cooling surface radiates to the surrounding surfaces mostly by the radiation. The highest radiant asymmetry is 1.88 °C near the south window side. Local thermal discomfort is not caused by the passive intelligent radiant cooling system in any position in the room because the limitation for asymmetric radiation to radiant cooling ceiling is 14 °C [20]–15 °C [17]. However, there are currently no recommendations or standards for asymmetric radiation of radiant floor cooling systems anywhere in the world.

### 4.4. The Cooling Capacity of the Passive Intelligent Radiant Cooling System

Figure 11 shows the cooling capacity of the passive intelligent radiant cooling system for one day.

The four supply and return pipes are connected in one distributor for four different rooms of the house boiler. The rooms are cooled automatically and intermittently only when the occupants use water while living at home. Therefore, the cooling capacity fluctuates because the cooling capacity calculation is fulfilled by Equation (5), which is influenced by the occupant's intermittent water usage. However, the room cooling surface temperature is relatively constant due to the high heat capacity of the concrete floor, which is put on as described in Figure 5. The cooling capacity for the radiant cooling system of room b was 25 W/m$^2$ while the occupant used water at home. This cooling capacity of the passive intelligent radiant cooling system is transferred mostly through radiation to the surrounding surfaces, including the occupants, because the air temperature near the radiant floor is similar to the floor surface temperature, as Figure 7 shows. Therefore, the operative temperature maintained a comfortable range for occupants during all cooling periods.

The previous study [15] showed a cooling capacity of 137 W/m² for one room compared to this study of 4 rooms. This suggests that the major difference between these two different systems is the amount of water flow rate.

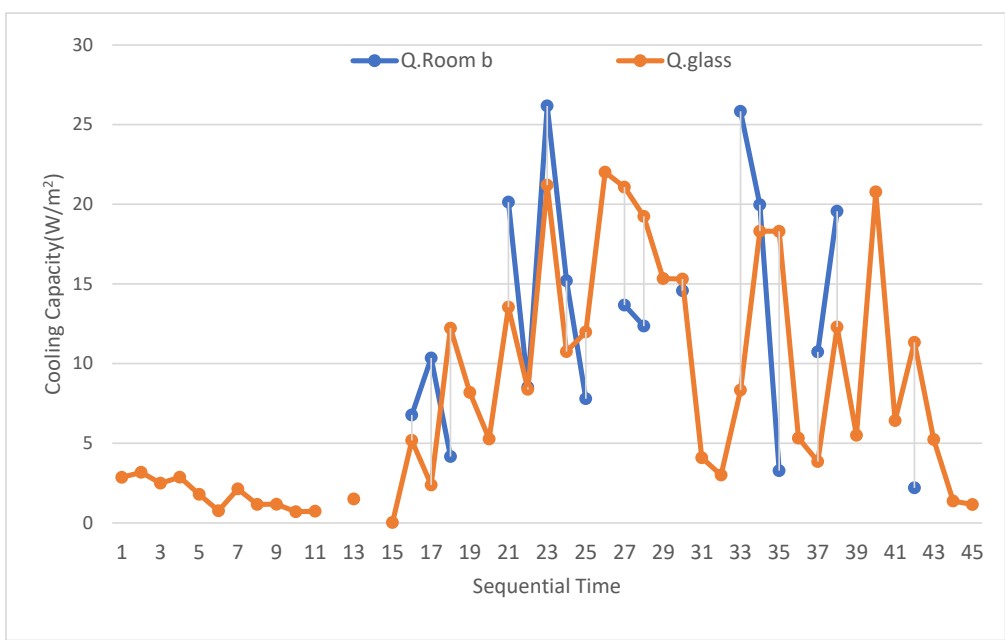

**Figure 11.** The cooling capacity of the passive intelligent radiant cooling system (7 August).

*4.5. Recommendations for an Energy-Efficient Dwelling with the Passive Intelligent Radiant Cooling System*

The author recommends the following considerations for an energy-efficient dwelling with a passive intelligent radiant cooling system. The radiant cooling system could meet the condensation problem on the radiant cooling surface depending on the thermal capacity of the system, surface temperature, water flow rate, and room humidity conditions. Therefore, outside shading is recommended to reduce the cooling load as much as possible, which can reduce the cooling load by up to 27–34% to prevent the possible condensation problem in Seoul, Korea [22]. A source flow ventilation system is recommended for the radiant floor cooling system to compensate for the system as per the radiant cooling floor surface temperature is similar to the 0.15 m height room air temperature so that the source flow ventilation system can be very effective.

**5. Conclusions**

Efficient cooling and heating solutions for nearly zero-energy solar dwellings are required to mitigate climate change and make more sustainable dwellings. The passive intelligent radiant cooling system has been converted from the radiant heating system over three years during the cooling period. The pipeline embedded in a radiant heating system, which is used for space heating just during the times when heating is necessary, can also be utilized to cool the room through enthalpy use of natural city water without any extra energy and power input by releasing natural city water through the embedded pipeline already installed for the radiant heating while the occupants are using the water at home. Natural city water used for radiant cooling can also be used for normal functions like flushing toilets, washing cars, using the washing machine, watering the garden, which correspond to approximately 56% of the water we use at home. As a result, the embedded pipes of the radiant heating system can be converted to a radiant cooling system with a minimum of added installation and control systems without any energy or power input.

No uncomfortable asymmetric radiation has been encountered for the cooling period, so the cooling spaces are well controlled within the comfortable cooling range without any

energy and power input during three years of operation. The passive intelligent radiant cooling concept by the enthalpy use of natural city water could be a nice solution for comfortable and reasonable zero-energy dwellings because no extra cooling energy and power are required to cool the space by the use of enthalpy and pressure from the natural city water.

The radiant cooling floor surface temperature is similar to air temperatures near the floor cooling surface, so there is little convective heat transfer. Alternatively, radiant cooling floor surfaces almost all radiate cold radiation flux to the ceiling surface, south wall surface, west wall surface, occupant, and window surface to reduce the surface temperature to be comfortable for the occupants. This passive intelligent radiant floor cooling concept automatically causes a fall in the surrounding surface temperatures, including 0.7 °C at the ceiling and 0.5 °C at the west wall against the room air temperature near the corresponding surface to increase the thermal comfort of the occupant just by the cold radiation from the radiant cooling floor surface. The average OT was 25.69 °C, which is a comfortable range for occupants even on the hottest summer day. The room OT range is maintained at 25.5–25.8 °C even on the hottest day.

The source flow ventilation system is recommended for the passive intelligent radiant cooling system to compensate for the system as per the radiant cooling floor surface temperature is similar to the 0.15 m height room air temperature so that the source flow ventilation system can be very effective.

In the future, the heat flow density on a passive intelligent radiant cooling floor surface will be more accurately evaluated in the experimental setup by the pipe spacing, thickness, heat conductivity of the layer above the pipe, water flow rate, etc.

**Funding:** This research received no external funding.

**Institutional Review Board Statement:** Not applicable.

**Informed Consent Statement:** Not applicable.

**Data Availability Statement:** Not applicable.

**Acknowledgments:** This research is supported by Sehan University.

**Conflicts of Interest:** The authors declare no conflict of interest.

## Nomenclature

| | |
|---|---|
| Asym | Asymmetric radiation [°C] |
| $A_{du}$ | Human body surface area [m$^2$] |
| h | Enthalpy [kJ/kg] |
| H | Height [cm] |
| m | Water flow rate [kG/s] |
| MRT | Mean radiant temperature [°C] |
| OT | Operative temperature [°C] |
| t | Temperature [°C] |
| T | Absolute temperature [°K] |
| W | Weight [Kg] |
| Ø | View factor [-] |
| Indices | |
| air | Air |
| coef | Coefficient |
| b | The human body |
| o | Opposite side |
| ot | Operative temperature |
| si | Room surface i, |
| 1 | Inlet position [-] |
| 2 | Outlet position [-] |
| u | Underside |

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
