# Peer review of "Thermal Behavior of Passive Intelligent Radiant Cooling Systems"

_processes, doi:10.3390/pr10122666_

Round 1

Reviewer 1 Report

Efficient cooling and heating systems for almost zero-energy solar houses are being studied as a means of preventing climate change and building sustainable home.

By releasing natural city water through the embedded pipeline already set up for the radiant heating, which is used for space heating only during the heating period, the installed pipeline for a radiant heating system, which is used for space heating only during the heating period, can also be used to cool the room simply by the enthalpy use of the natural city water without any energy and power input. The paper provides valuable information on the numerical and experimental results in buildings. However, you ought to place more emphasis on the novelty of your work; in my opinion, neither the abstract nor the introduction provide sufficient information to determine whether the study is novel or not. I feel that this paper is suitable for publication after major revisions as suggested below.

1.   The abstract has to be revised because it does not adequately describe the most noteworthy outcomes of their work.

2.   The introduction does not list articles where the approaches and models have been applied, nor does it specify which of the researchers' notable findings are included. You might draw attention to the originality of your research by reworking the introduction.

3.   The authors list the benefits of employing the models, but they never mention their shortcomings or how doing so would impact the outcomes.

4.   It needs to provide a more thorough explanation of the mathematical model, governing equations, applicable assumptions and convergence criteria was employed.

5.   Some equations are not clear.

Author Response

Thank you for your kind review.

Reviewer 2 Report

The study is interesting. However, the whole expression is unclear, and the focus is not prominent. The structure is unreasonable. I cannot recommend it to be published in its current state. The followings provide some comments for further study.

1)      What are the innovation and motivation of this study? Please explain and present them in detail.

2)      The introduction section is too general. It is suggested that the author adequately cite previous studies and highlight the motivation of the study.

3)      Inadequate references were used in the introduction. The author should select appropriate references and highlight the research features according to the content of the study. Please develop the literature.

4)      Due to the problems mentioned above in the introduction, this section should be completely restructured.

5)      For the statement line (65) to line (70) please include the evidence

6)      Please provide detailed information for pipes used in this study

7)      For statement line (126).  32 thermal points have been measured by the Data logger per every 30 or 60 minutes…. Why not just 30 or 60 minutes? In this case, inconsistencies occur in the results.

8)      Provide detailed information about the equipment used in this study (e.g., accuracy, model, country)

9)      I could not see the meaning of some abbreviations in the equations in the article. Please provide a nomenclature.

10)  The study contains no discussion. Differences or similarities of findings from previous studies should be explained.

11)  Figures in the article can be drawn better.

12)  Conclusions should be appropriately simplified and highlight the conclusions and instructive results of the study.

13)  The manuscript needs extensive revision for language and grammar.

 Although the subject of the article is interesting, there are significant problems in conveying its focus, interpreting it, and highlighting its novelty. 

Author Response

 Thank you for your kind review.

Reviewer 3 Report

Advances in the domain have been achieved and the content of the study is related to the scope of the journal but the author should consider the below comments to improve the paper.

1.                   The author needs to revise the abstract because in the current form it is more likely the succinct description of a review article than of a research paper. It is necessary to clearly formulate the aim of the paper and give some insight on the particularities of the study as well as on the main findings.

2.                   Introduction section needs to be consolidated and modified. There are a low number of references (only 9) and there little (numerical) information about how other studies manages to resolve the same problem. What is their advantage, disadvantage, performance in comparison to the solution proposed in the current study? It is necessary to clearly indicate what the novelty of the current study is and what is its importance for the readers of the journal?

3.                   In my opinion the title and content of section 2.1 needs slight reformulation. First of all I think that or energy or power needs to be deleted from the title. Both terms are referring to the same aspect. Secondly, stating that cooling can be achieved "without any energy and power input" is not quite accurate. I am sure that the author refers to unnecessary use of extra pumps for fluid transportation (requiring extra energy input), which partially is true. But, if the pipe system becomes longer (plus additional hydraulic resistances), by flowing the water through the embedded pipeline already installed for the conventional radiant heating system, than pressure drop and implicitly power consumption will increase. So, it would be better to state that it is a significant reduction of power consumption if we compare this case study with the one that uses a different thermal agent than city water.

4.                   In line 68 the author states that " water is supplied at least 90cm under the earth in Korea. " then at line 70 we read "Korean water supply facilities should be buried at least under 90cm of earth. " So which is it? It is supplied at that depth or it should be?

5.                   Please insert a list of abbreviations which would be useful considering the notations used in the paper. It is enough to define abbreviations only once in the text do not define them repeatedly.

6.                   Conclusions section is too long. Needs to formulated more shortly and with more numerical data

7.                   The cooling capacity of the passive intelligent radiant cooling system needs to be compared with the one of other system defined in the literature.

8.                   I recommend that instead of "X" use a different symbol for the operator in equations 1-5.

Author Response

Thank you for your kind review.

Round 2

Reviewer 1 Report

Accept in present form

Author Response

Thank you for your kind review.

Reviewer 2 Report

The author has improved the article, taking into account the suggestions of the reviewers. However, there are still some corrections to the article that have not been resolved.

Line 57: The author stated that ‘The number of articles investigating radiative cooling structures for building applications has increased rapidly in recent years’. Which articles? Information about these articles should be given before this sentence. Thus, the introduction section can be literarily enriched.

A lot of spelling and grammatical errors are still in the article. For example:

Line 119: the units of "M" should be changed to "m".

Line 213-126: I can not understand the numbers 21, 34, and 40 (inside the brackets).

Line 210-216: Please write "a.m." and "p.m." in lowercase

Figure 9-10 should be redrawn. x1 and y1 are overlap. Please correct the title of the y-axis (South…)

The author sometimes used the operative temperature and sometimes the abbreviation "OT". Please correct. Use the full form of the word the first time it appears in the article (Line 156), and then abbreviate it thereafter.

Figure 11. Please change the y-axis title "asymmetric radiation" to "asymmetric radiation temperature (°C) "

Figure 12. The unit should be Wm-2 or W/m2 (not W/m^2).

The article as well as the Figures contain too many grammatical errors. Therefore, the reviewer strongly suggests that the article be linguistically revised, and the Figures redesigned.

Author Response

The author has improved the article, taking into account the suggestions of the reviewers. However, there are still some corrections to the article that have not been resolved.

Line 57: The author stated that ‘The number of articles investigating radiative cooling structures for building applications has increased rapidly in recent years’. Which articles? Information about these articles should be given before this sentence. Thus, the introduction section can be literarily enriched.

->I added the corresponding literature [11].

The number of articles investigating radiative cooling structures for building applications has increased rapidly in recent years[11]’.

A lot of spelling and grammatical errors are still in the article. For example:

Line 119: the units of "M" should be changed to "m".

Revised “M” to “m” in figure 5

Line 213-126: I can not understand the numbers 21, 34, and 40 (inside the brackets).

->Added sequential time (ST) in figure 5 and revised in the corresponding text.

Line 210-216: Please write "a.m." and "p.m." in lowercase

Corrected.

Figure 9-10 should be redrawn. x1 and y1 are overlap. Please correct the title of the y-axis (South…)

Figure 9 is redrawn. And it is difficult to redraw figure 10 for the overlapped x and y, but the y-axis is fine and I think that the readers can read from figure 9.

The author sometimes used the operative temperature and sometimes the abbreviation "OT". Please correct. Use the full form of the word the first time it appears in the article (Line 156), and then abbreviate it thereafter.

Corrected.

Figure 11. Please change the y-axis title "asymmetric radiation" to "asymmetric radiation temperature (°C) "

Corrected.

Figure 12. The unit should be Wm-2 or W/m2 (not W/m^2).

 Corrected.

The article as well as the Figures contain too many grammatical errors. Therefore, the reviewer strongly suggests that the article be linguistically revised, and the Figures redesigned.

Thank you very much for your thorough review.

Reviewer 3 Report

The author made significant progress with the improvement of the paper. I have only on observation that was not considered in the previous step. Please use or power or energy and do not use both words in the same sentence. They refer to the same technical aspect.

Author Response

The author made significant progress with the improvement of the paper. I have only on observation that was not considered in the previous step. Please use or power or energy and do not use both words in the same sentence. They refer to the same technical aspect.

->

I removed “power or energy” in unnecessary places, especially in the abstract.

But I think a few places should stay there including one in the abstract because the system doesn’t need any energy for cooling and power for the water supply.

Thank you for your thorough review!